# Salt-inducible kinases mediate nutrient-sensing to link dietary sugar and tumorigenesis in *Drosophila*

**Susumu Hirabayashi[1,2]\*, Ross L Cagan[3]**

[1]Metabolism and Cell Growth Group, MRC Clinical Sciences Centre, Imperial College London, London, United Kingdom; [2]PRESTO, Japan Science and Technology Agency, Kawaguchi, Japan; [3]Department of Developmental and Regenerative Biology, Icahn School of Medicine at Mount Sinai, New York, United States

**Abstract** Cancer cells demand excessive nutrients to support their proliferation but how cancer cells sense and promote growth in the nutrient favorable conditions remain incompletely understood. Epidemiological studies have indicated that obesity is a risk factor for various types of cancers. Feeding *Drosophila* a high dietary sugar was previously demonstrated to not only direct metabolic defects including obesity and organismal insulin resistance, but also transform Ras/Src-activated cells into aggressive tumors. Here we demonstrate that Ras/Src-activated cells are sensitive to perturbations in the Hippo signaling pathway. We provide evidence that nutritional cues activate Salt-inducible kinase, leading to Hippo pathway downregulation in Ras/Src-activated cells. The result is Yorkie-dependent increase in Wingless signaling, a key mediator that promotes diet-enhanced Ras/Src-tumorigenesis in an otherwise insulin-resistant environment. Through this mechanism, Ras/Src-activated cells are positioned to efficiently respond to nutritional signals and ensure tumor growth upon nutrient rich condition including obesity.

**\*For correspondence:** susumu.hirabayashi@csc.mrc.ac.uk

**Competing interests:** The authors declare that no competing interests exist.

## Introduction

The prevalence of obesity is increasing globally. Obesity impacts whole-body homeostasis and is a risk factor for severe health complications including type 2 diabetes and cardiovascular disease. Accumulating epidemiological evidence indicates that obesity also leads to elevated risk of developing several types of cancers (*Calle et al., 2003*; *Renehan et al., 2008*; *Arnold et al., 2014*). However, the mechanisms that link obesity and cancer remain incompletely understood. Using *Drosophila*, we recently developed a whole-animal model system to study the link between diet-induced obesity and cancer and provided a potential explanation for how obese and insulin resistant animals are at increased risk for tumor progression (*Hirabayashi et al., 2013*).

*Drosophila* fed a diet containing high levels of sucrose (high dietary sucrose or 'HDS') developed sugar-dependent metabolic defects including accumulation of fat (obesity), organismal insulin resistance, hyperglycemia, hyperinsulinemia, heart defects and liver (fat body) dysfunctions (*Musselman et al., 2011*, *2013*; *Na et al., 2013*; *Na et al., 2015*). Inducing activation of oncogenic Ras and Src together in the *Drosophila* eye epithelia led to development of small benign tumors within the eye epithelia. Feeding animals HDS transformed Ras/Src-activated cells from benign tumor growths to aggressive tumor overgrowth with tumors spread into other regions of the body (*Hirabayashi et al., 2013*). While most tissues of animals fed HDS displayed insulin resistance, Ras/Src-activated tumors retained insulin pathway sensitivity and exhibited an increased ability to import glucose. This is reflected by increased expression of the Insulin Receptor (InR), which was activated through an increase in canonical Wingless (Wg)/dWnt signaling that resulted in evasion of diet-mediated insulin

**eLife digest** Around the world, obesity has become a much more common condition. It is a serious health concern, which can increase a person's risk of developing type 2 diabetes, heart disease and certain types of cancer. People who develop type 2 diabetes become insensitive to a hormone called insulin. This hormone normally helps the body to process sugar, and so insensitivity to insulin causes excess sugar to build up in the blood. The excess sugar may provide the extra nutrients cancer cells need to grow.

In 2013, researchers fed a high sugar diet to fruit flies that had been genetically engineered to develop eye tumors to study how obesity caused by a high sugar diet affects tumor growth. The high sugar diet caused the tumors to grow more aggressively. This happened because normal cells became insensitive to insulin, but the tumor cells didn't. This allowed the tumor cells to use the extra sugar to fuel their growth. The experiments showed that the tumor cells had more insulin receptors than normal cells because a molecular switch that controls the receptors was turned on. But it wasn't exactly clear how the cancer genes and excess sugar flipped that switch.

Now, Hirabayashi and Cagan—who were both involved in the 2013 work—show that together cancer genes and excess sugar turn on a protein in the flies that senses sugar. This protein, called Salt-inducible kinase, blocks a cellular mechanism that normally limits the growth of cells. With this check on cellular growth blocked, the molecular switch that boosts the number of insulin receptor turns on. This in turn allows the excess sugar to fuel rapid growth of the tumor. In this way, tumor cells know when the sugars are available and make sure they grow in a nutrient-rich condition such as obesity. In the future, scientists may use this new information to develop treatments that help stop the growth of obesity-linked tumors. But first it must be confirmed whether excess sugar and cancer genes behave the same way in humans.

resistance in Ras/Src-activated cells. Conversely, expressing a constitutively active isoform of the Insulin Receptor in Ras/Src-activated cells (InR/Ras/Src) was sufficient to elevate Wg signaling, promoting tumor overgrowth in animals fed a control diet. These results revealed a circuit with a feed-forward mechanism that directs elevated Wg signaling and InR expression specifically in Ras/Src-activated cells. Through this circuit, mitogenic effects of insulin are not only preserved but are enhanced in Ras/Src-activated cells in the presence of organismal insulin resistance.

These studies provide an outline for a new mechanism by which tumors evade insulin resistance, but several questions remain: (i) how Ras/Src-activated cells sense the organism's increased insulin levels, (ii) how nutrient availability is converted into growth signals, and (iii) the trigger for increased Wg protein levels, a key mediator that promotes evasion of insulin resistance and enhanced Ras/Src-tumorigenesis consequent to HDS. In this manuscript, we identify the Hippo pathway effector Yorkie (Yki) as a primary source of increased Wg expression in diet-enhanced Ras/Src-tumors. We demonstrate that Ras/Src-activated cells are sensitized to Hippo signaling, and even a mild perturbation in upstream Hippo pathway is sufficient to dominantly promote Ras/Src-tumor growth. We provide functional evidence that increased insulin signaling promotes Salt-inducible kinases (SIKs) activity in Ras/Src-activated cells, revealing a SIKs-Yki-Wg axis as a key mediator of diet-enhanced Ras/Src-tumorigenesis. Through this pathway, Hippo-sensitized Ras/Src-activated cells are positioned to efficiently respond to insulin signals and promote tumor overgrowth. These mechanisms act as a feed-forward cassette that promotes tumor progression in dietary rich conditions, evading an otherwise insulin resistant state.

## Results

### Yorkie mediates increased Wg expression in diet-enhanced Ras/Src-tumors

Ras/Src tumors were generated in the developing *Drosophila* eye epithelium by pairing targeted expression of the activated *dRas1* isoform $ras1^{G12V}$ with targeted knockout (*Lee and Luo, 1999*) of the negative regulator of Src, C-terminal src kinase ($csk^{-/-}$). Feeding animals a diet containing 1.0 M sucrose (high dietary sucrose or 'HDS') transformed these Ras/Src-activated cells from benign growths

to aggressive tumors associated with emergent tumor spread to other parts of the body (*Figure 1A,B*) (*Hirabayashi et al., 2013*). In HDS-fed animals, Ras/Src-activated cells promoted gene expression of InR through increased canonical Wg-dependent signaling, leading to increased insulin sensitivity in Ras/Src-activated cells in otherwise insulin resistant animals. Expression of a constitutive active isoform of Insulin Receptor (*inr^CA*) in Ras/Src-activated cells (*inr ^CA,ras1^G12V;csk^−/−*) was sufficient to promote elevation of Wg levels and tumor growth even in a control diet, establishing an InR-Wg-InR amplification circuit that promotes aggressive tumorigenesis (*Figure 1E*) (*Hirabayashi et al., 2013*).

Reducing Wg by RNAi (*wg^RNAi*) did not affect normal eye tissue growth of the late third instar larvae (*Figure 1—figure supplement 1*). However, reducing Wg in Ras/Src-activated cells (*ras1^G12V;csk^−/−, wg^RNAi*) fed HDS or in InR/Ras/Src-activated cells (*inr ^CA,ras1^G12V;csk^−/−,wg^RNAi*) fed a control diet significantly suppressed tumor growth (*Figure 1C,F*). As a result, whereas only 35.9% of *ras1^G12V;csk^−/−* animals in HDS and 42.6% of *inr^CA,ras1^G12V;csk^−/−* animals in control diet initiated pupariation, most *ras1^G12V;csk^−/−,wg^RNAi* animals and *inr^CA,ras1^G12V;csk^−/−,wg^RNAi* animals successfully pupariated (*Figure 1H*). These observations identify Wg as an essential mediator of diet-enhanced Ras/Src-tumors or InR/Ras/Src-tumors. However, the factors that elevate Wg expression in diet- or InR-activated Ras/Src-tumors has not been identified.

The Hippo pathway is an evolutionarily conserved signaling pathway that regulates tissue growth and cell fate (*Harvey and Tapon, 2007*; *Halder and Johnson, 2011*). The Hippo pathway regulates growth through the transcriptional co-activator Yki, a *Drosophila* homolog of mammalian YAP/TAZ (*Huang et al., 2005*). The core pathway kinase effector Warts (Wts) phosphorylates Yki and inhibits its activity by sequestering Yki in the cytoplasm (*Dong et al., 2007*; *Zhao et al., 2007*; *Oh and Irvine, 2008*). Conversely, loss of components in the core Hippo complex results in translocation of Yki into the nucleus where it regulates factors that promote proliferation and inhibit cell death (*Goulev et al., 2008*; *Wu et al., 2008*; *Zhang et al., 2008*).

Yki activation has been previously associated with increased expression of Wg (*Cho et al., 2006*). Inhibition of Yki activity by over-expressing Wts led to small clones (*Figure 1—figure supplement 1*). Similarly, over-expression of Wts in Ras/Src-activated cells (*ras1^G12V;csk^−/−,wts*) fed HDS or in InR/Ras/Src-activated cells (*inr^CA,ras1^G12V;csk^−/−,wts*) fed a control diet led to a strong suppression of tumor growth and animal lethality (*Figure 1D,G and H*). Importantly, increased Wg expression was lost in these clones, indicating that Yki is required for the increased Wg expression observed in diet- or InR-activated Ras/Src-tumors (*Figure 1I–L*).

## Yorkie target genes are upregulated in diet-enhanced Ras/Src-tumors

Myc and cyclin E are well-established transcriptional targets of Yki in *Drosophila* (*Tapon et al., 2002*; *Udan et al., 2003*; *Neto-Silva et al., 2010*). Myc and cyclin E were strongly elevated in eye clones of *ras1^G12V;csk^−/−* animals fed HDS and in *inr^CA,ras1^G12V;csk^−/−* animals fed a control diet (*Figure 1—figure supplement 2*) compared to controls. As previously reported, *diap1* gene expression—assessed by the Yki transcriptional reporter *diap1-lacZ*—was strongly increased in the *ras1^G12V;csk^−/−* clones of animals raised in HDS compared to animals fed a control diet ((*Hirabayashi et al., 2013*) *Figure 2A,B*). Upon closer examination, *diap1* gene expression was at most mildly increased in most *ras1^G12V;csk^−/−* clones in animals fed a control diet (*Figure 2A*, arrowheads). This increase in Yki activity was not sufficient to promote tumor overgrowth: *ras1^G12V;csk^−/−* clones of animals raised in a control diet were progressively eliminated from the tissue by apoptotic cell death (*Hirabayashi et al., 2013*). Activation of insulin signaling pathway alone (*inr^CA*) failed to elevate *diap1* expression in a control diet (*Figure 2C*). However, the triple combination (*inr ^CA,ras1^G12V;csk^−/−*) led to strongly elevated *diap1* gene expression, including in animals fed a control diet (*Figure 2D*). These results indicate that activation of Yki is an emergent property of Ras and Src co-activation, and increased insulin signaling further promotes Yki activity in Ras/Src-activated cells.

## Ras/src-activated cells are sensitized to upstream Hippo signals

The FERM domain protein Expanded (Ex) is both an upstream regulator of the Hippo pathway and a transcriptional target of Yki, forming a negative feedback loop (*Hamaratoglu et al., 2006*). An enhancer trap fly line in which a *lacZ* gene is inserted in the *ex* locus (*ex^674*; [*Boedigheimer and Laughon, 1993*]) can therefore be used to reduce *ex* activity as well as a readout of Yki transcriptional activity. Removing a functional genomic copy of *ex* (*ex^+/−*) did not affect normal eye tissue growth

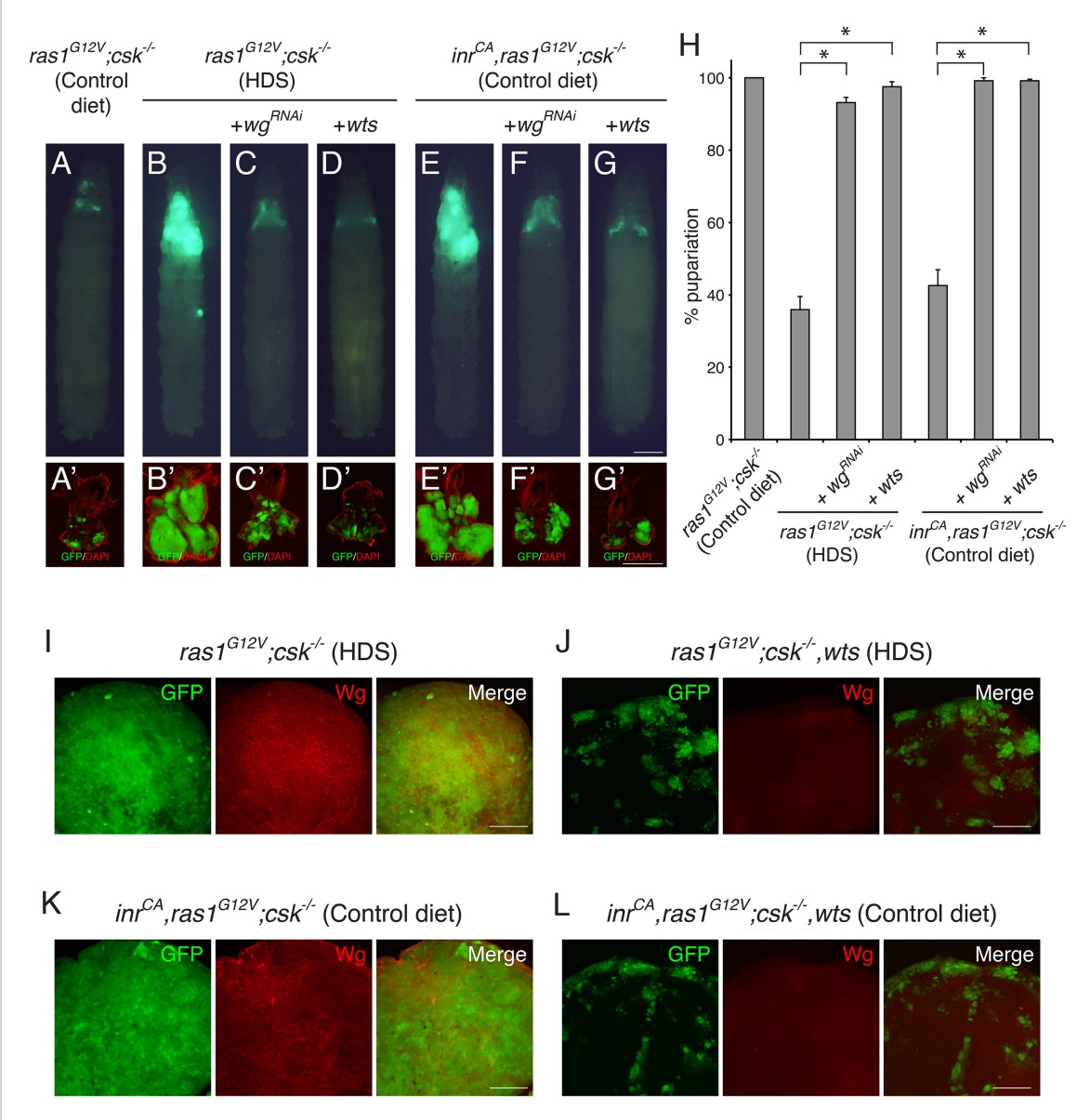

**Figure 1**. Yorkie Activity is Required for Increased Wg Expression in Diet-enhanced Ras/Src-tumors. (**A–G**) Developmental stage matched third instar larvae with the genotype, (**A, B**) $ras1^{G12V};csk^{-/-}$, (**C**) $ras1^{G12V};csk^{-/-},wg^{RNAi}$, (**D**) $ras1^{G12V};csk^{-/-},wts$, (**E**) $inr^{CA},ras1^{G12V};csk^{-/-}$, (**F**) $inr^{CA},ras1^{G12V};csk^{-/-},wg^{RNAi}$, and (**G**) $inr^{CA},ras1^{G12V};csk^{-/-},wts$, raised on indicated diets. Images were taken at the same magnification. Scale bar, 500 μm. (**A'–G'**) Matching dissected eye epithelial tissue stained with DAPI (red). Images were taken at the same magnification. Scale bar, 500 μm. (**H**) Percent pupariation of animals from indicated genotypes and diets. Column bars represent the mean of three independent experiments. Error bars denote s.e.m. Total *n* was 166, 431, 309, 291, 204, 200, and 251 from left to right. Asterisks indicate statistically significant difference (*p < 0.01 t-test). Numerical data are available in *Figure 1—source data 1*. (**I–L**) Wg staining (red) of eye tissue from (**I**) $ras1^{G12V};csk^{-/-}$, (**J**) $ras1^{G12V};csk^{-/-},wts$, (**K**) $inr^{CA},ras1^{G12V};csk^{-/-}$, and (**L**) $inr^{CA},ras1^{G12V};csk^{-/-},wts$ animals raised on indicated diets. Scale bars, 50 μm.

The following source data and figure supplements are available for figure 1:

**Source data 1**. Percent pupariation of animals from indicated genotypes and diets.

**Figure supplement 1**. Effect of reducing Wg or over-expressing Wts in the eye tissue.

**Figure supplement 2**. Yorkie target genes are upregulated in diet-enhanced Ras/Src-tumors.

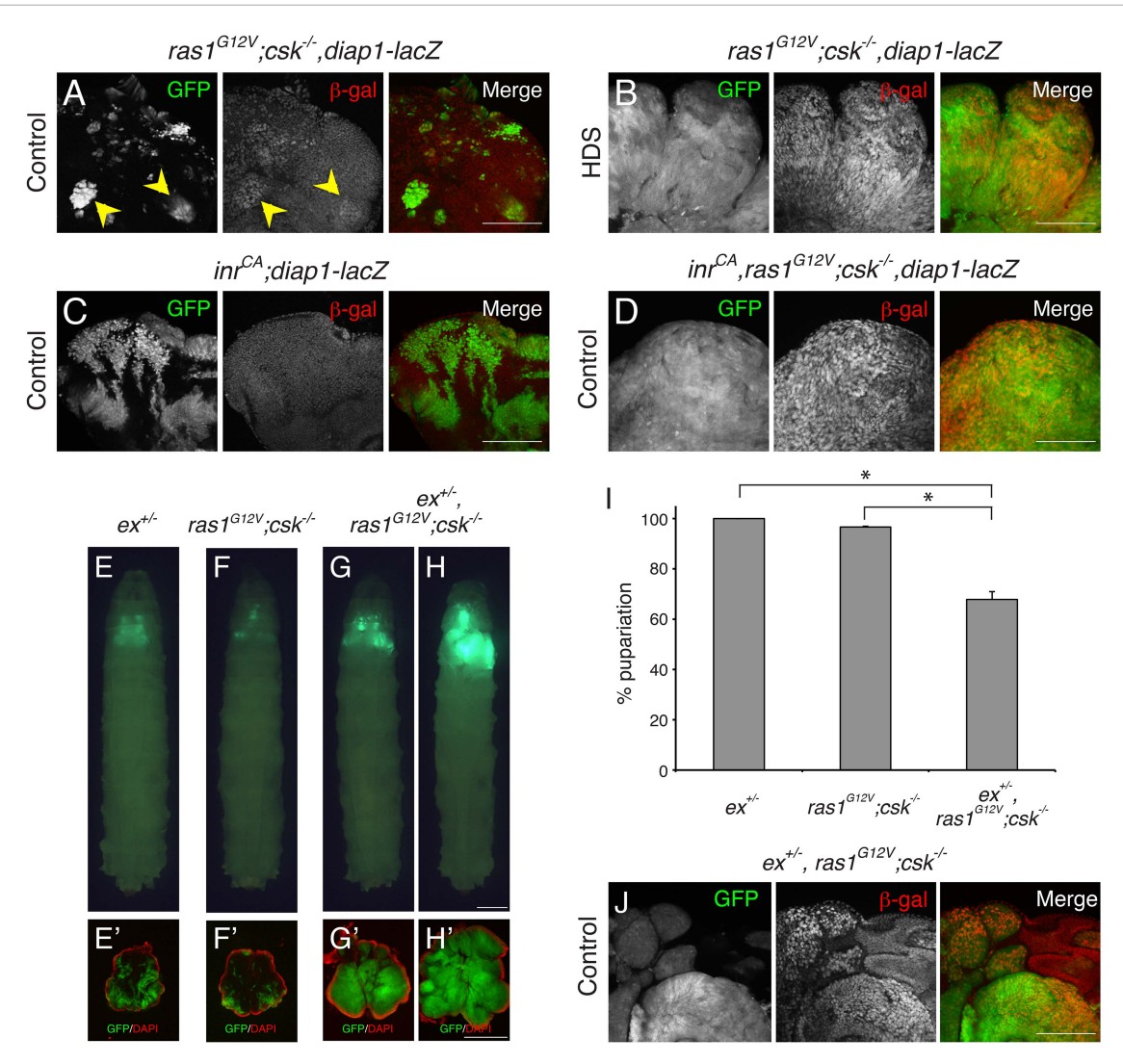

**Figure 2**. Ras/Src-activated Cells are Sensitive to Perturbations in the Hippo Signaling. (**A–D**) β-galactosidase (β-gal) staining (red) of eye tissue from (**A**, **B**) *ras1^{G12V};csk^{−/−},diap1-lacZ*, (**C**) *inr^{CA};diap1-lacZ*, (**D**) *inr^{CA},ras1^{G12V};csk^{−/−},diap1-lacZ* animals raised on indicated diets. Scale bars, 50 μm. (**E–H**) Developmental stage matched third instar larvae raised on control diet with the genotype, (**E**) *ex^{+/−}*, (**F**) *ras1^{G12V};csk^{−/−}*, (**G**, **H**) *ex^{+/−},ras1^{G12V};csk^{−/−}*. Images were taken at the same magnification. Scale bar, 500 μm. (**E'–H'**) Matching dissected eye epithelial tissue stained with DAPI (red). Images were taken at the same magnification. Scale bar, 500 μm. (**I**) Percent pupariation of animals from indicated genotypes. Column bars represent the mean of three independent experiments. Error bars denote s.e.m. Total *n* of 389, 238, and 206 from left to right. Asterisks indicate statistically significant difference (*p < 0.01 t-test). Numerical data are available in *Figure 2—source data 1* (**J**) β-galactosidase (β-gal) staining (red) of *ex^{+/−}, ras1^{G12V};csk^{−/−}* animals raised on control diet. Scale bar, 50 μm.

The following source data is available for figure 2:

**Source data 1**. Percent pupariation of animals from indicated genotypes.

(*Figure 2E*). Surprisingly, reducing *ex* in *ras1^{G12V};csk^{−/−}* animals (*ex^{+/−},ras1^{G12V};csk^{−/−}*) dominantly promoted growth of Ras/Src-activated cells in a control diet (*Figure 2F-H*). As a consequence, 40% of *ex^{+/−},ras1^{G12V};csk^{−/−}* animals in control diet failed to initiate pupariation, dying as larvae with overgrown eye tissue (*Figure 2I*). Immunostaining using anti-β-galactosidase antibody indicated that *ex* gene expression was strongly increased in *ex^{+/−}, ras1^{G12V};csk^{−/−}* clones of animals raised in a control diet, demonstrating Yki activation in these clones (*Figure 2J*). Together these results provide compelling evidence that Ras/Src-activated cells are functionally linked to Hippo pathway activity, as

even a subtle perturbation of upstream Hippo signaling is sufficient to dominantly promote tumor Ras/Src-tumor overgrowth.

## Warts kinase activity is downregulated in diet-enhanced Ras/Src-tumors

To determine whether diet-enhanced Ras/Src-tumors promote Yki activity through the core Hippo signaling pathway, we performed Western-blot analysis using an antibody to the phosphorylated Serine-168 residue of Yki, a standard indication of Wts kinase activity. As anticipated, phosphorylation of Yki was significantly reduced in genotypically $wts^{-/-}$ eye tissues. Dietary sucrose did not alter Yki phosphorylation in *lacZ*-expressing control clones (*Figure 3A*). In $ras1^{G12V};csk^{-/-}$ animals, however, HDS led to a reduction in Yki phosphorylation to a level comparable to loss of *wts* (*Figure 3A*). Phosphorylation was similarly reduced in eye tissues of $inr^{CA},ras1^{G12V};csk^{-/-}$ animals fed a control diet, indicating that increased insulin signaling is sufficient to suppress Wts kinase activity in Ras/Src-activated cells (*Figure 3A*). We did not observe significant changes in total Yki levels (*Figure 3A*). Our results indicate that (i) Ras/Src-tumors in the presence of HDS or (ii) InR/Ras/Src-tumors in a control diet promote Yki activity through inhibition of Wts kinase activity.

## Salt-inducible kinases mediate diet-enhanced ras/src-tumor overgrowth

Salt-inducible kinases (SIKs) were recently shown to regulate wing tissue growth through Hippo pathway activity in *Drosophila* (*Wehr et al., 2013*). Phosphorylation of Salvador (Sav) at Serine-413 by SIK led to dissociation of the Hippo complex and activation of Yki (*Wehr et al., 2013*). In the eye tissue of $ras1^{G12V};csk^{-/-}$ larvae fed HDS, Serine-413 phosphorylation of Sav was strongly upregulated (*Figure 3A*). Similarly, phosphorylation of Sav was strongly elevated in $inr^{CA},ras1^{G12V};csk^{-/-}$ animals fed a control diet, demonstrating that increased insulin signaling in Ras/Src-activated cells promotes SIK activity (*Figure 3A*).

Previous studies in mammals and *Drosophila* have shown that SIKs are activated by Akt (*Dentin et al., 2007*; *Wang et al., 2011*; *Choi et al., 2015*). Reducing Akt activity through a hypomorphic allele of *akt* ($ras1^{G12V};csk^{-/-},akt^{hypo/hypo}$) suppressed Ras/Src-tumor growth and reduced phosphorylation of Sav in animals raised on HDS (*Figure 3—figure supplement 1*). Similarly, tumor growth and phosphorylation of Sav was suppressed in $inr^{CA},ras1^{G12V};csk^{-/-},akt^{hypo/hypo}$ animals fed a control diet (*Figure 3—figure supplement 1*). These results demonstrate that—in $ras1^{G12V};csk^{-/-}$ larval eye tissue fed HDS and $inr^{CA},ras1^{G12V};csk^{-/-}$ animals fed a control diet—activation of SIKs are mediated by Akt.

To examine whether the SIKs are required for diet-enhanced Ras/Src-tumorigenesis, we used a transgenic RNA-interference line that targets both *sik2* and *sik3* transcripts for knockdown (*Wehr et al., 2013*). Reducing SIK2/3 in $ras1^{G12V};csk^{-/-}$ animals ($sik2^{RNAi},ras1^{G12V};csk^{-/-}$) suppressed diet-enhanced Ras/Src-tumor growth (*Figure 3B,C*). Importantly, reducing SIK2/3 by itself did not significantly affect normal eye tissue growth of animals fed HDS (*Figure 3—figure supplement 2*), indicating that SIK2/3 is functionally required for Ras/Src-tumor growth in the presence of HDS. Feeding HG-9-91-01, a potent inhibitor of SIKs (*Clark et al., 2012*), led to suppression of tumor growth and animal lethality in both $ras1^{G12V};csk^{-/-}$ animals fed HDS and in $inr^{CA},ras1^{G12V};csk^{-/-}$ animals fed a control diet (*Figure 3F-J*). We conclude that activation of SIKs is functionally required for diet-enhanced Ras/Src-tumorigenesis.

Conversely, expression of a constitutive active isoform of SIK2 ($sik2^{CA}$ (*Wehr et al., 2013*)) in Ras/Src-activated cells ($ras1^{G12V};csk^{-/-},sik2^{CA}$) was sufficient to promote Ras/Src-dependent tumor overgrowth even in a control diet (*Figure 4A,B*). Western blot analysis confirmed increased phosphorylation of Sav and reduced phosphorylation of Yki in $ras1^{G12V};csk^{-/-},sik2^{CA}$ tumors (*Figure 4C*). Wg expression was strongly upregulated in $ras1^{G12V};csk^{-/-},sik2^{CA}$ tumors in animals raised on a control diet (*Figure 4D,E*), further linking SIKs to Hippo pathway activity. Taken together, these results demonstrate that SIKs provide the upstream Hippo signal that mediates Ras/Src-tumorigenesis in diet-induced obese animals.

## Discussion

We previously demonstrated that Ras/Src-activated cells preserve mitogenic effects of insulin under the systemic insulin resistance induced by HDS-feeding of *Drosophila* (*Hirabayashi et al., 2013*). Evasion of insulin resistance in Ras/Src-activated cells is a consequence of a Wg-dependent increase in InR gene expression (*Hirabayashi et al., 2013*). In this study, we identify the Hippo pathway effector

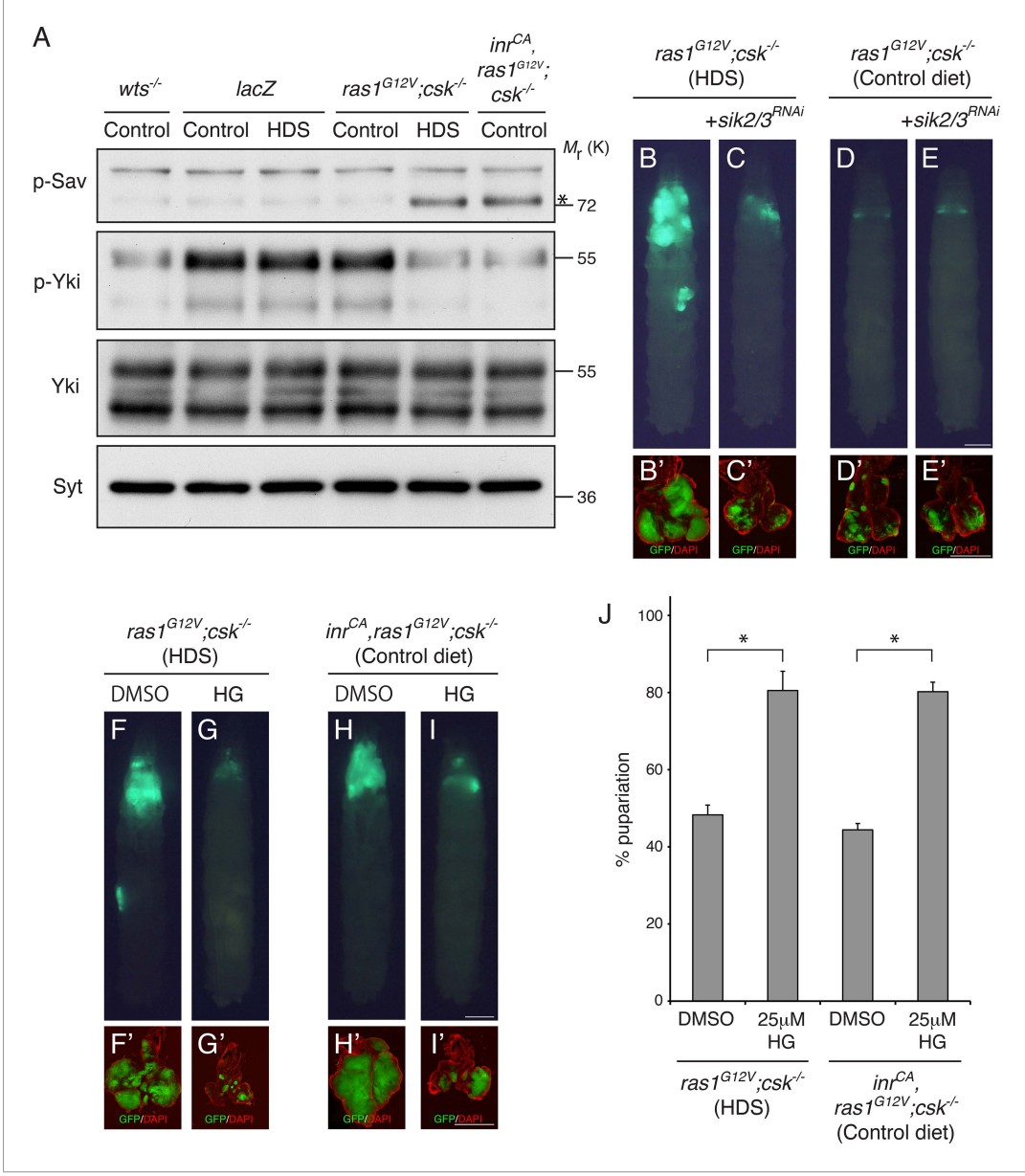

**Figure 3**. Salt-inducible Kinases are Required for Diet-enhanced Ras/Src-tumorigenesis. (**A**) Extracts from dissected eye tissues of third instar larvae were examined by immunoblotting using antibodies against phospho-Sav (p-Sav; * indicates p-Sav specific band; the upper band is a non-specific band showed as an internal loading control), phospho-Yki (p-Yki), total Yki (Yki), and Syntaxin (Syt). (**B**, **C**) Developmental stage matched third instar larvae raised on HDS with the genotype, (**B**) $ras1^{G12V};csk^{-/-}$, and (**C**) $sik2/3^{RNAi},ras1^{G12V};csk^{-/-}$. (**D**, **E**) Developmental stage matched third instar larvae raised on control diet with the genotype, (**D**) $ras1^{G12V};csk^{-/-}$, and (**E**) $sik2/3^{RNAi},ras1^{G12V};csk^{-/-}$. Images were taken at the same magnification. Scale bar, 500 μm. (**F**, **G**) $ras1^{G12V};csk^{-/-}$ animals raised on HDS containing (**F**) 0.05% DMSO, or (**G**) 25 μM HG-9-91-01. (**H**, **I**) $inr^{CA},ras1^{G12V};csk^{-/-}$ animals raised on control diet containing (**H**) 0.05% DMSO, or (**I**) 25 μM HG-9-91-01. Images were taken at the same magnification. Scale bar, 500 μm. (**B'**–**I'**) Matching dissected eye epithelial tissue stained with DAPI (red). Images were taken at the same magnification. Scale bar, 500 μm. (**J**) Percent pupariation of DMSO or HG-9-91-01 treated animals from indicated genotypes and diets. Column bars represent the mean of three independent experiments. Error bars denote s.e.m. Total $n$ of 139, 76, 123, and 72 from left to right. Asterisks indicate statistically significant difference (*p < 0.01 t-test). Numerical data are available in *Figure 3—source data 1*.

The following source data and figure supplements are available for figure 3:

**Source data 1**. Percent pupariation of animals from indicated genotypes and diets.

*Figure 3. continued on next page*

*Figure 3. Continued*

**Figure supplement 1**. Akt mediates activation of SIKs in Ras/Src-tumors.
**Figure supplement 2**. Reducing SIK2/3 by RNAi did not affect normal eye tissue growth.

Yki as a primary source of the Wnt ortholog Wg in diet-enhanced Ras/Src-tumors. Mechanistically, we provide functional evidence that activation of SIKs promotes Yki-dependent Wg-activation and reveal a SIK-Yki-Wg-InR axis as a key feed-forward signaling pathway that underlies evasion of insulin resistance and promotion of tumor growth in diet-enhanced Ras/Src-tumors (*Figure 4F*).

In animals fed a control diet, we observed at most a mild increase in Yki reporter activity within $ras1^{G12V};csk^{-/-}$ cells (*Figure 2A*). A previous report indicates that activation of oncogenic Ras ($ras1^{G12V}$) led to slight activation of Yki in eye tissue (*Ohsawa et al., 2012*; *Enomoto and Igaki, 2013*; *Enomoto et al., 2015*). Activation of Src through over-expression of the *Drosophila* Src ortholog Src64B has been shown to induce autonomous and non-autonomous activation of Yki (*Enomoto and Igaki, 2013*). In contrast, inducing activation of Src through loss of *csk* ($csk^{-/-}$) failed to elevate *diap1* expression (data not shown). Our results indicate that activation of Yki is an emergent property of activating Ras plus Src ($ras1^{G12V};csk^{-/-}$). However, this level of Yki-activation was not sufficient to promote stable tumor growth of Ras/Src-activated cells in the context of a control diet: Ras/Src-activated cells were progressively eliminated from the eye tissue (*Hirabayashi et al., 2013*). It was, however, sufficient to sensitize Ras/Src-activated cells to upstream Hippo pathway signals: loss of a genetic copy of *ex*—which was not sufficient to promote growth by itself—dominantly promoted tumor growth of Ras/Src-activated cells even in animals fed a control diet (*Figure 2G-I*). These data provide compelling evidence that Ras/Src-transformed cells are sensitive to upstream Hippo signals.

SIK was recently demonstrated to phosphorylate Sav at Serine-413, resulting in dissociation of the Hippo complex and activation of Yki (*Wehr et al., 2013*). SIKs are required for diet-enhanced Ras/Src-tumor growth in HDS (*Figure 3C*). Conversely, expression of a constitutively activated isoform of SIK was sufficient to promote Ras/Src-tumor overgrowth even in a control diet (*Figure 4B*). Mammalian SIKs are regulated by glucose and by insulin signaling (*Wang et al., 2008*, *2011*). However, a more recent report indicated that glucagon but not insulin regulates SIK2 activity in the liver (*Patel et al., 2014*). Our data demonstrate that increased insulin signaling is sufficient to promote SIK activity through Akt in Ras/Src-activated cells (*Figure 3A*, *Figure 3—figure supplement 1*). We conclude that SIKs couple nutrient (insulin) availability to Yki-mediated evasion of insulin resistance and tumor growth, ensuring Ras/Src-tumor growth under nutrient favorable conditions.

Our results place SIKs as key sensors of nutrient and energy availability in Ras/Src-tumors through increased insulin signaling and, hence, increased glucose availability. SIK activity promotes Ras/Src-activated cells to efficiently respond to upstream Hippo signals, ensuring tumor overgrowth in organisms that are otherwise insulin resistant. One interesting question is whether this mechanism is relevant beyond the context of an obesity-cancer connection: both Ras and Src have pleiotropic effects on developmental processes including survival, proliferation, morphogenesis, differentiation, and invasion, and these mechanisms may facilitate these processes under nutrient favorable conditions. From a treatment perspective our data highlight SIKs as potential therapeutic targets. Limiting SIK activity through compounds such as HG-9-91-01 may break the connection between oncogenes and diet, targeting key aspects of tumor progression that are enhanced in obese individuals.

## Materials and methods

### Fly stocks

$UAS\text{-}ras1^{G12V}$, $UAS\text{-}inr^{A1325D}$ ($inr^{CA}$), $UAS\text{-}wts$, $ex^{697}$($ex\text{-}lacZ$) , $Diap1^{j5C8}$ ($diap1\text{-}lacZ$), $akt^{04226}$ flies were obtained from the Bloomington *Drosophila* Stock Center. $UAS\text{-}wg^{RNAi}$ and $UAS\text{-}sik2^{RNAi}$ flies were obtained from Vienna *Drosophila* RNAi Center. The following stocks were kindly provided to us: $FRT82B$, $csk^{Q156Stop}$ by A. O'Reilly and M. Simon; $ey(3.5)\text{-}FLP1$ by G. Halder; $UAS\text{-}sik2^{S1032A}$ ($sik2^{CA}$) by N. Tapon; $wts^{X1}$ by C. Pfleger.

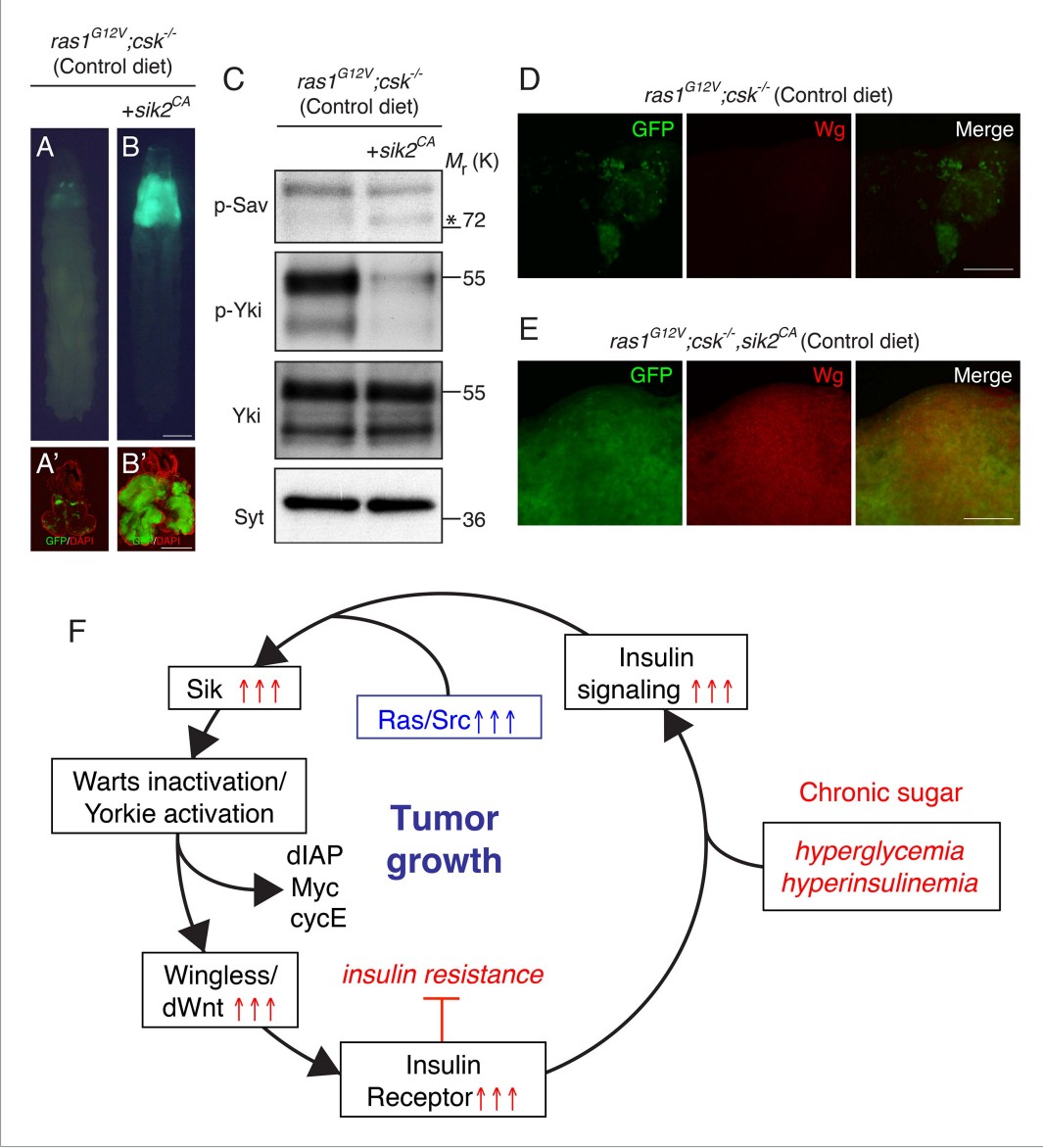

**Figure 4**. Activation of Salt-inducible Kinase Promotes Ras/Src-tumor Growth. (**A**, **B**) Developmental stage matched third instar larvae raised on control diet with the genotype, (**A**) *ras1^{G12V};csk^{−/−}*, and (**B**) *ras1^{G12V};csk^{−/−},sik2^{CA}*. Images were taken at the same magnification. Scale bar, 500 µm. (**A'**, **B'**) Matching dissected eye epithelial tissue stained with DAPI (red). Images were taken at the same magnification. Scale bar, 500 µm. (**C**) Extracts from dissected eye tissues of *ras1^{G12V};csk^{−/−}* and *ras1^{G12V};csk^{−/−},sik2^{CA}* animals fed a control diet were examined by immunoblotting using antibodies against phospho-Sav (p-Sav; * indicates p-Sav specific band; the upper band is a non-specific band showed as an internal loading control), phospho-Yki (p-Yki), total Yki (Yki), and Syntaxin (Syt). (**D**, **E**) Wg staining (red) of eye tissue from (**D**) *ras1^{G12V};csk^{−/−}*, and (**E**) *ras1^{G12V};csk^{−/−},sik2^{CA}* animals raised on control diet. Scale bars, 50 µm. (**F**) Model of diet-enhanced tumorigenesis of Ras/Src-activated cells.

To create eyeless-driven green fluorescent protein (GFP)–labeled clones, flies with the genotype *ey (3.5)-FLP1; act > y+>gal4,UAS-GFP; FRT82B,tub-gal80* were crossed with flies with the following genotypes: (a) *UAS-ras1^{G12V}; FRT82B, csk^{Q156Stop}/TM6b*; (b) *UAS-ras1^{G12V}; FRT82B, csk^{Q156Stop}, UAS-wg^{RNAi}/TM6b*; (c) *UAS-ras1^{G12V}; FRT82B, csk^{Q156Stop}, UAS-wts/TM6b*; (d) *UAS-inr^{A1325D}, UAS-ras1^{G12V}; FRT82B, csk^{Q156Stop}/TM6b*; (e) *UAS-inr^{A1325D}, UAS-ras1^{G12V}; FRT82B, csk^{Q156Stop}, UAS-wg^{RNAi}/TM6b*; (f) *UAS-inr^{A1325D}, UAS-ras1^{G12V}; FRT82B, csk^{Q156Stop}, UAS-wts/TM6b*; (g) *FRT82B, UAS-wg^{RNAi}*; (h) *FRT82B, UAS-wts*; (i) *UAS-ras1^{G12V}; diap1-lacZ, FRT82B, csk^{Q156Stop}/TM6b*; (j) *UAS-inr^{A1325D}, UAS-ras1^{G12V};*

*diap1-lacZ, FRT82B, csk$^{Q156Stop}$/TM6b*; (k) *UAS-inr$^{A1325D}$; diap1-lacZ, FRT82B/TM6b*; (l) *ex$^{697}$; FRT82B/ SM6-TM6b*; (m) *ex$^{697}$, UAS-ras1$^{G12V}$; FRT82B, csk$^{Q156Stop}$/SM6-TM6b*; (n) *FRT82B, wts$^{X1}$/TM6b*; (o) *UAS-lacZ; FRT82B*; (p) *UAS-sik2$^{RNAi}$, UAS-ras1$^{G12V}$; FRT82B, csk$^{Q156Stop}$/TM6b*; (q) *UAS-ras1$^{G12V}$; UAS-sik2$^{CA}$, FRT82B, csk$^{Q156Stop}$/TM6b*; (r) *UAS-sik2$^{RNAi}$; FRT82B*; (s) *UAS-ras1$^{G12V}$; FRT82B, csk$^{Q156Stop}$, akt$^{04226}$ /TM6b*; (t) *UAS-inr$^{A1325D}$, UAS-ras1$^{G12V}$; FRT82B, csk$^{Q156Stop}$, akt$^{04226}$ /TM6b*.

## Cultures

Cultures were carried out on Bloomington semi-defined medium (described by the Bloomington *Drosophila* stock center) with modifications. Detailed recipes for control diet and HDS is previously described (*Musselman et al., 2011*). The following final concentrations of carbohydrates were included: 0.15 M sucrose (control diet) and 1.0 M sucrose (HDS). Cultures were performed at 25°C.

## Immunofluorescence, western blotting

These procedures were performed as previously described (*Hirabayashi et al., 2013*). Primary antibodies used for immunofluorescence were: mouse anti-Wingless (DSHB: Developmental Studies Hybridoma Bank, Iowa City, IA, United States), mouse anti-Cyclin E (DSHB), mouse anti-β-galactosidase (DSHB), rabbit anti-Myc (Santa Cruz Biotechnology, Dallas, TX, United States). Western blots were probed with antibodies against Yki (gift from K Irvine) (*Oh and Irvine, 2008*), phospho-Yki (pS168) (gift from D. *Pan*) (*Dong et al., 2007*), phospho-Sav (pS413) (gift from N Tapon) (*Wehr et al., 2013*), and Syntaxin (DSHB).

## Drugs

HG-9-91-01 (MedChem Express, Princeton, NJ, United States) was solubilized in DMSO and diluted directly into the fly medium and vortexed extensively to obtain a homogeneous culture.

## Acknowledgements

We thank C Pfleger, N Tapon, and members of Cagan, Pfleger and Tapon laboratories for helpful discussions, and C Pfleger, A O'Reilly, M Simon, D Pan, K Irvine, and N Tapon for kindly providing reagents. We also thank the Bloomington *Drosophila* Stock Center (NIH P40OD018537), Vienna RNAi Stock Center and Developmental Studies Hybridoma Bank for fly strains and antibodies. This research was supported by grants from the National Cancer Institute (R01CA170495, R01-CA109730; R L C), PRESTO, Japan Science and Technology Agency (SH) and intramural funding from Medical Research Council (SH).

## Additional information

### Funding

| Funder | Grant reference | Author |
|---|---|---|
| National Cancer Institute (NCI) | R01CA170495 | Ross L Cagan |
| National Cancer Institute (NCI) | R01CA109730 | Ross L Cagan |
| Japan Science and Technology Agency (JST) | PRESTO | Susumu Hirabayashi |
| Medical Research Council (MRC) | Intramural Funding | Susumu Hirabayashi |

The funders had no role in study design, data collection and interpretation, or the decision to submit the work for publication.

### Author contributions

SH, Conception and design, Acquisition of data, Analysis and interpretation of data, Drafting or revising the article; RLC, Conception and design, Analysis and interpretation of data, Drafting or revising the article

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
