## [Decision Letter]

Thank you for submitting your work entitled “Salt-Inducible Kinases Mediate Nutrient-Sensing To Link Dietary Sugar and Tumorigenesis in *Drosophila”* for peer review at *eLife*. Your submission has been favorably evaluated by Tony Hunter (Senior Editor) and two reviewers, one of whom is a member of our Board of Reviewing Editors.

The reviewers have discussed the reviews with one another and the Reviewing editor has drafted this decision to help you prepare a revised submission.

The work addresses the link between diabetes/obesity and increased tumour susceptibility. In previous work, the authors have established a *Drosophila* model in which a combination of dietary glucose and Ras + Src oncogenes can lead to enhanced tumorigenesis by inducing the expression of Wnt and Insulin Receptor. In the current manuscript, the authors identify the Hippo pathway effector Yorkie as an essential mediator of Wnt induction in the tumors. Mechanistically, the authors show that activation of Yki is due to activation of Salt Inducible Kinases (SIKs) by Ras + Src + dietary glucose. Overall, the study is well conducted and clearly presented, and will be of broad interest to the cancer, diabetes and developmental growth control communities. The authors have identified a potentially important mechanism by which nutrients cooperate oncogenes to drive tumorigenesis. Furthermore, this represents the first clear demonstration that the Hippo pathway is regulated by metabolism.

Essential revisions:

1) Some additional insight into how SIK is activated in response to HDS would strengthen the study. Is it Akt-dependent, as is suggested by the strong activation of Akt by HDS in their previous study (Hirabayashi et al., Cell 2013)? This could be addressed by activating Akt in normal diet or RNAi depletion of Akt in HDS.

2) Please include statistics (Reviewer 1), the molecular weight markers and scale bars (Reviewer 2) in the revision.

---

## [Author Response]

Essential revisions:

1) Some additional insight into how SIK is activated in response to HDS would strengthen the study. Is it Akt-dependent, as is suggested by the strong activation of Akt by HDS in their previous study (Hirabayashi et al., Cell 2013)? This could be addressed by activating Akt in normal diet or RNAi depletion of Akt in HDS.

Previous studies in mammals and *Drosophila* have shown that SIKs are activated by Akt. To assess the role of Akt in activation of SIKs in our model, we used a hypomorphic allele of *akt* (*akt*^*04226*^) and established *ras1*^*G12V*^*;csk*^*−/−*^,*akt*^*hypo/hypo*^ and *inr*^*CA*^*,ras1*^*G12V*^*;csk*^*−/−*^,*akt*^*hypo/hypo*^ lines. Reducing Akt-activity in *ras1*^*G12V*^*;csk*^*−/−*^ animals raised on a HDS or in *inr*^*CA*^*,ras1*^*G12V*^*;csk*^*−/−*^ animals raised on a control diet both suppressed tumor growth and activation of SIKs as assessed by Western-blot analysis using anti-phospho Sav antibody (Figure 3—figure supplement 1). Therefore, together with the evidence that increased insulin signaling in Ras/Src-activated cells (*inr*^*CA*^*,ras1*^*G12V*^*;csk*^*−/−*^) promotes SIK activity (Figure 3), we now clearly demonstrate that the activation of SIKs in the eye tissues of *ras1*^*G12V*^*;csk*^*−/−*^ larvae fed HDS and in *inr*^*CA*^*, ras1*^*G12V*^*;csk*^*−/−*^ animals fed a control diet are mediated by Akt.

2) Please include statistics (Reviewer 1), the molecular weight markers and scale bars (Reviewer 2) in the revision.

We have provided statistics for the quantification in Figures 1, 2 and 3, molecular weight markers in all Western-blot data and scale bars in all immunofluorescence data.